# SEMANTIC EMBEDDING-DRIVEN TOPOLOGY EXTRACTION WITH CORRELATION LAG AWARENESS

## ABSTRACT

Many application scenarios involve anomaly detection of multivariate time series, which exhibit complicated dependencies across different dimensions governed by physical laws or system design. Though such cross-dimensional dependencies usually serve as important references for human experts to detect anomalies from monitoring data, the algorithms in this domain have not explored and leveraged them thoroughly. On the one hand, many algorithms incorporating graph networks to model the cross-dimensional dependencies require establishing the topology manually. However, in many real-world application scenarios, there are usually thousands of indicators, such as complex IoT systems, aircraft control systems, and so on. Constructing such topologies manually is laborious, as the complexity of defining them grows quadratically with the number of dimensions. On the other hand, graph networks usually assume fixed and instantaneous dependencies, but we observe pronounced cross-dimensional correlation lag in complex system monitoring data, indicating that dependencies across dimensions are not static but dynamically shift with time intervals. To address these issues, we propose an **A**nomaly **D**etection Method Capturing **Se**mantic Topology and **C**orrelation lag (ADSec), which extracts topology from expert documents and monitoring data automatically and successfully models the cross-dimensional correlation lag. Specifically, ADSec extracts a semantic topology from expert documents and refines it with monitoring data. Besides, it leverages a novel Neural Hawkes process to model the cross-dimensional correlation lag and adjust the topology dynamically. Extensive experiments on four real-world datasets demonstrate that ADSec improves F1 Score by 5.8% averagely on multivariate time series with complex inter-dimension dependencies, compared with SOTA anomaly detection methods.

## 1 INTRODUCTION

In many real-world application scenarios, such as anomaly detection in aircraft control systems, there exist complex cross-dimensional dependencies. For instance, these include the voting logic in redundant channels (Davis, 1987) or the physical relationships between commanded force and control surface deflection (Nelson et al., 1998). Such dependencies are typically rooted in system design principles and physical laws. While they often provide valuable references for human experts when detecting anomalies in monitoring data, existing anomaly detection algorithms have not fully exploited this information.

MSCRED (Zhang et al., 2019) employs a signature matrix to model cross-dimensional correlations, and SARAD (Dai et al., 2024) trains a transformer to capture spatial correlations. However, these approaches are purely data-driven and fail to effectively incorporate domain knowledge about such dependencies. Many spatio-temporal anomaly detection methods (Yi et al., 2023; Mercatali et al., 2024; Huang et al., 2023) adopt graph networks to represent cross-dimensional correlations, but these approaches require manually constructing the topology. In practice, scenarios such as industrial IoT systems and aircraft control systems often involve thousands of monitoring indicators, making manual topology construction impractical given its quadratic complexity with respect to the number of dimensions.

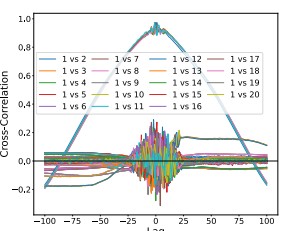 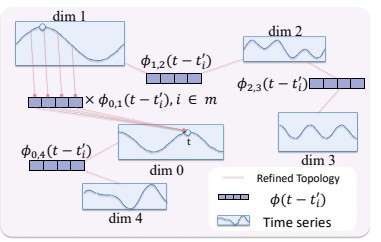 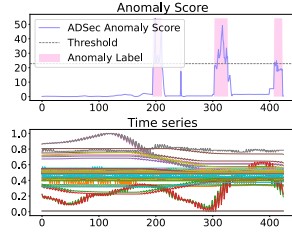

(a) Cross dimension correlation lag

(b) Illustration of Neural Hawkes

(c) Model performance

Figure 1: (a) The figure shows how the correlation coefficients between different dimensions of a high-dimensional time series vary with lag. In the legend, "1 vs 2" denotes the cross-correlation between the first and second dimensions; (b) The figure illustrates the working mechanism of the Neural Hawkes process and explains why it can adapt to temporal scenarios where the topology varies across different time steps; (c) The figure presents the anomaly scores produced by ADSec alongside the corresponding time series and anomaly labels, demonstrating the effectiveness of ADSec.

Moreover, graph-based approaches typically assume fixed and instantaneous dependencies across dimensions. In reality, variations in one dimension often induce changes in another only after a certain delay. For example, in aircraft control systems, the rudder angle responds several time steps after a steering wheel command force is applied. As illustrated in Fig. 1(a), where we plot the cross-correlation functions of monitoring data from an aircraft control system, the peak values across most dimensions do not appear at lag 0. This observation indicates that cross-dimensional dependencies vary with time intervals and exhibit cross-dimensional correlation lags and using a static and instantaneous topology could not precisely model such a lag.

To effectively leverage expert knowledge of cross-dimensional dependencies and address the limitations of graph networks in handling dynamic dependencies with correlation lag, we propose an **A**nomaly **D**etection method capturing **Se**mantic Topology and **C**orrelation lag (ADSec). Specifically, ADSec first constructs a semantic topology from a large language model (LLM) fine-tuned on expert documents. This semantic topology is then refined to better align with monitoring data by a refinement module. In addition, to capture dynamic cross-dimensional dependencies with temporal lag, we propose a novel Neural Hawkes process that learns impact functions $\phi(t - t_i')$ for each time lag $t_i'$ based on the refined topology, as illustrated in Fig. 1(b).

The main contributions of this paper are summarized in the following:

- We propose a framework that integrates semantic topology derived from professional documents with data-pattern topology extracted from monitoring data.

- We design a semantic topology extraction module that effectively leverages diverse professional documents, even when they contain inconsistent terminologies.

- We propose a novel Neural Hawkes Process to model dynamically evolving dependencies with correlation lags across different monitoring dimensions.

- We conduct extensive experiments demonstrating that ADSec delivers robust and consistent improvements in anomaly detection performance as shown in Fig. 1(c), outperforming the strongest state-of-the-art methods across multiple datasets.

## 2 PRELIMINARY

### 2.1 MULTIVARIATE HAWKES PROCESS

A multivariate Hawkes process (Cai et al., 2022) models how the occurrence of an event in one dimension exerts a continuous and dynamically evolving influence on the same and other dimensions over future time, governed by an impact function. Specifically, when an event occurs in a given dimension, it increases the conditional intensities of events in the same and other dimensions, with the magnitude and duration of the influence modulated by the form of the impact function. This allows the Hawkes process to capture intricate, time-variant interactions across multiple dimensions.

Formally, for an event set $\mathcal{V}$, given empirical event rates $E \in \mathbb{R}^{T \times |\mathcal{V}|}$, $E[t, i]$ represents the rate of event $i$ at time slot $t$. The conditional intensity function $\lambda_k(t|\mathcal{H})$ for event $k$ at $t^{th}$ time slot is given by Eq. 1, where $T^-$ is a set of time slots that are before the time slot $t$, $\phi_{i,k}$ is an impact function characterizing the time-variant influence between event $i$ and $k$, $\mu_k$ represents the spontaneous occurrence rate of event $k$, and $\mathcal{H}$ is the observing event history at $t^{th}$ time slot.

$$\lambda_k(t|\mathcal{H}) = \mu_k + \sum_{i \in \mathcal{V}} \sum_{t' \in T^-} \phi_{i,k}(t - t')E[t', i] \tag{1}$$

## 2.2 PROBLEM SETUP

Given a multivariate time series $X \in \mathbb{R}^{T \times n}$ with semantically meaningful names for each dimension, where $X[i] \in \mathbb{R}^n$ denotes the time series value at $i^{th}$ time slot, the ground truth of whether each time slot is anomaly is denoted by $y \in \mathbb{R}^T$, where $y[i] = 1$ denotes the $i^{th}$ time slot is an anomaly and $y[i] = 0$ denotes it is not. We divide the time series $X$ by sliding window and use $X_i$ to denote the $i^{th}$ window. We use $[\vec{a}_i]_{i=1}^n$ to denote the word embeddings for each dimension name in LLM. The semantic topology between different dimensions are represented by an adjacency matrix $A \in \mathbb{R}^{n \times n}$. The data-driven relation pattern between different dimensions are denoted by an adjacency matrix $S \in \mathbb{R}^{n \times n}$. The reconstruction-based anomaly detection methods obtain a reconstructed series $\hat{X}$ and use the reconstruction error $\left|X[i] - \hat{X}[i]\right|$ as the anomaly score $M[i]$ for $i^{th}$ time slot. The reconstruction-based anomaly detection methods aim to minimize objective function shown in Eq. 2, where $\mathbf{1}$ denotes the indicative function and $\mathcal{T}$ denotes the threshold found by the reconstruction-based anomaly detection methods.

$$\sum_{i=1}^T |y[i] - \mathbf{1}(M[i] - \mathcal{T})| \tag{2}$$

## 3 METHODOLOGY

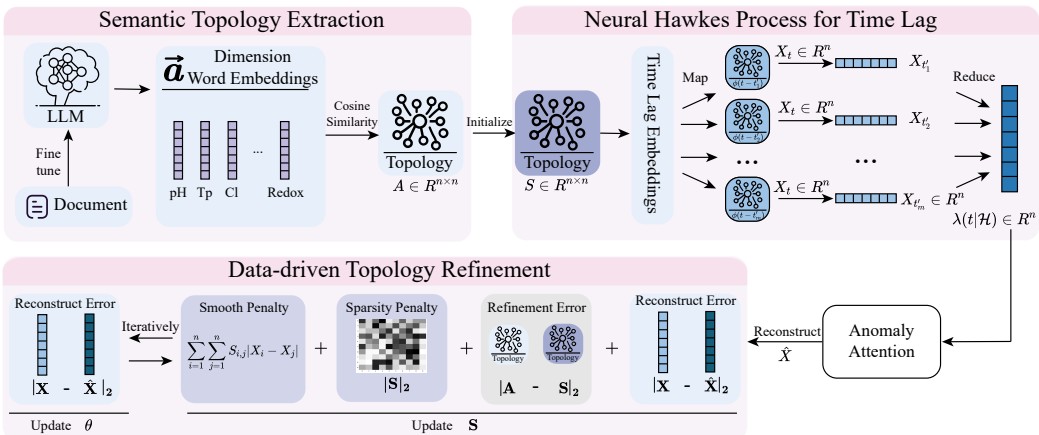

Figure 2: The model architecture of ADSec

## 3.1 OVERVIEW

The architecture of ADSec is shown in Fig. 2. We firstly use the professional document to fine-tune an LLM and obtain the word embeddings for each dimension name. Based on these word embedding we construct the semantic topology. Subsequently, to make the semantic topology more consistent with the actual data, in the training process, we refine the semantic topology by a designed loss function. In order to model the cross dimensional correlation lag, we use Time Lag Embeddings to obtain impact functions of Neural Hawkes process for different temporary lag based on refined topology. After that, we use anomaly attention mechanism (Xu et al., 2022) to reconstruct original time series from the conditional intensity obtained from Neural Hawkes process. The topology refinement and time series reconstruction is trained iteratively.

## 3.2 Semantic Topology Extraction

To better leverage the expert knowledge, we extract semantic topology relation between different dimensions in the dataset from professional documents. However, even within the same organization, it is highly likely that different departments use varying expressions to refer to the same monitoring indicator in their respective domain-specific documents. Traditional knowledge graph extraction methods tend to create separate entities for the same object when it is referred to by different names, leading to inconsistency and redundancy in the resulting graph (Getoor & Machanavajjhala, 2012). To mitigate the potential disruption that inconsistent terminology may introduce when extracting the semantic topological relationships among different dimensions, we extract the semantic topological structure among dimensions based on their word embeddings, allowing the analysis to focus on the underlying semantics rather than superficial differences in the naming of dimensions. We firstly fine-tune the embedding layer of a causal self-attention-based large language model (LLM) (Radford et al., 2019) using the Low-Rank Adaptation (LoRA) technique (Hu et al., 2022), with domain-specific documents and a text completion objective. This fine-tuning process aligns the model's word embeddings with the semantics of the target domain, enabling the LLM to better capture and represent expert knowledge specific to the current field. The fine-tuning details can be found in Appendix. A.5. Subsequently, we extract the word embeddings corresponding to each dimension name in the current dataset. If a dimension name is tokenized into multiple tokens, we compute its final embedding by averaging the embeddings of all associated tokens. We use $\vec{a}_i$ to denote the embeddings of $i^{th}$ dimension.

We observe that after fine-tuning the model on domain-specific documents, the angles between the word embeddings reflect expert knowledge regarding the relationships between dimensions. Specifically, when two dimensions are positively correlated according to domain expertise, the angle between their embeddings tends to be smaller, resulting in a higher cosine similarity. Conversely, when two dimensions are negatively correlated, the angle between their embeddings tends to be larger, leading to a lower (often negative) cosine similarity. For example, water temperature and flow velocity are known to have a positive correlation; after fine-tuning, the cosine similarity between their embeddings is positive and greater than that between unrelated dimensions. In contrast, the concentration of chlorinated compounds and water temperature are known to exhibit a negative correlation; correspondingly, after fine-tuning, the cosine similarity between their embeddings becomes negative and smaller than that observed between unrelated dimensions. We discuss this finding in Section 4.5 experimentally.

Based on the aforementioned findings, we model the semantic topological relation between dimensions by calculating the cosine similarity between their embeddings. We use $A \in \mathbb{R}^{n \times n}$ to denote the semantic topology adjacency matrix, where $A[i][j]$ denotes the semantic relation between $i^{th}$ and $j^{th}$ dimension, as defined in Eq. 3.

$$A[i][j] = \frac{\vec{a}_i \cdot \vec{a}_j}{|\vec{a}_i| \, |\vec{a}_j|} \tag{3}$$

This approach enables us to overcome the problem of using different terminologies referring to the same object, thereby effectively integrating expert knowledge from multiple documents that adopt distinct terminological systems. For example, different documents use the terms "steering wheel control force" and "steering wheel command force" to describe the same aircraft monitoring indicator, respectively. These two terms are tokenized into ['ste', 'ering', 'Ġwheel', 'Ġcontrol', 'Ġforce'] and ['ste', 'ering', 'Ġwheel', 'Ġcommand', 'Ġforce'], respectively. Through this tokenization, different documents collaboratively fine-tune the embeddings of shared subword tokens such as 'ste', 'ering', 'Ġwheel', and 'Ġforce'. Consequently, when extracting the final word representations, the model can effectively fuse expert knowledge about the same concept from different documents.

## 3.3 Neural Hawkes Process for Time Lag

In traditional graph neural networks, the modeling of topological dependencies across different dimensions typically assumes static and instantaneous relationships. However, this assumption fails to capture the fact that inter-dimensional dependencies may evolve dynamically over time and often exhibit temporal lag. For example, consider a scenario where dimension $\mathcal{A}$ represents changes in control stick force in an aircraft, while dimension $\mathcal{B}$ represents changes in the deflection angle of the aircraft control surface. In this case, changes in $\mathcal{A}$ do not immediately lead to changes in $\mathcal{B}$;

instead, the influence of $\mathcal{A}$ on $\mathcal{B}$ becomes evident only after several time steps. To accurately capture such temporally lagged and dynamically evolving dependencies, we adopt the Hawkes process to model the inter-dimensional topological relations over time. We construct impact functions across different temporal intervals based on the underlying topology and optimize them through data-driven training. Specifically, we first initialize a trainable adjacency matrix $S$ using semantic topology, which encodes the structural prior between dimensions. Taking the $t^{th}$ sliding window $X_t$ as an example, let $T^- = \{t'_1, t'_2, \ldots, t'_m\}$ denote the set of time slot immediately preceding $t$. The impact function is given by Eq. 4, where $W_i$ are trainable parameters, $\phi_0$ is the instantaneous impact function, and $\phi(t - t'_i)$ is the impact function when time interval is equal to $t - t'_i$.

$$\phi_0 = SW_0, \ \phi(t - t'_i) = SW_i, \ i \in [1, m] \tag{4}$$

Consequently, we model the conditional intensity function by Eq. 5, where $\mu \in \mathbb{R}^n$ is trainable spontaneous response rate, the term $\sum_{i=1}^{m} X_{t'_i} \cdot \phi(t - t'_i)$ models the time-lagged influence both within and across dimensions, and the term $X_t \cdot \phi_0$ models the instantaneous impact both within and across dimensions. After obtaining the conditional intensity function, we use Anomaly Attention mechanism (Xu et al., 2022) to reconstruct the original time series from it. The reconstruction time series is denoted by $\hat{X}$.

$$\lambda(t|\mathcal{H}) = \mu + \sum_{i=1}^{m} X_{t'_i} \cdot \phi(t - t'_i) + X_t \cdot \phi_0 \tag{5}$$

In standard Hawkes processes, the impact function is typically required to satisfy two conditions: non-negativity and finite integrability. However, in ADSec, we do not impose the non-negativity constraint. This is because, in many complex real-world scenarios, the relationship between dimensions can be suppressive (negatively correlated) rather than excitatory (positively correlated). As such, allowing the impact function to take negative values enables ADSec to capture negative correlation between dimensions. Regarding finite integrability, this property still holds in our setting. Since the set $T^-$ is finite and the values of the impact function $\phi$ within this set are bounded, the integration of impact function remains finite.

## 3.4 Data-driven Topology Refinement

The semantic topology extracted from expert documents does not fully align with the underlying cross-dimensional correlation in the dataset. Therefore, we use data-driven method to refine the semantic topology extracted from professional document. We denote the refined topology by $S$, which is initialized with the semantic topology $A$. We adopt an alternating optimization strategy: we first fit the topology structure $S$ and fix the remaining model parameters to minimize a designed loss function $\mathcal{L}_1$; then, with the topology structure fixed, we update the remaining model parameters $\theta$ by minimizing the reconstruction error $\mathcal{L}_2$. These two steps are iteratively performed until convergence.

In the first step, inspired by (Jin et al., 2020), we optimize the refined topology $S$ with the objective of enhancing the smoothness of the graph neural network and reducing the reconstruction error. To this end, we incorporate a sparsity regularization term, as well as a penalty term that constrains the deviation of the optimized $S$ from the original topology extracted from professional documents. The overall loss function used in this step is defined in Eq. 6. In the second step, we optimize $\theta$ by the reconstruction error, as shown in Eq. 7.

$$\mathcal{L}_1(S) = \|X - \hat{X}\|_2 + \gamma_1 \sum_{i=1}^{n} \sum_{j=1}^{n} S_{i,j} \|X[:, i] - X[:, j]\|_2 + \gamma_2 \|A - S\|_2 + \gamma_3 \|S\|_2 \tag{6}$$

$$\mathcal{L}_2(\theta) = \|X - \hat{X}\|_2 \tag{7}$$

## 4 Experiment

We have made extensive experiments on four datasets and make the following contributions:

- ADSec can achieve the best F1 score across the four datasets compared with SOTA, only requiring reasonable time and memory overhead increase.

- ADSec is insensitive to the hyperparameters.
- Every module in ADSec contributes to its final performance. Especially, the semantical topology can successfully capture important correlation between dimensions and the neural Hawkes process can precisely capture cross dimensional correlation lag.

## 4.1 EXPERIMENT SETUP

**Baselines.** We choose the SOTA anomaly detection methods, Anomaly Transformer (Xu et al., 2022), TranAD (Tuli et al., 2022), AutoFormer (Wu et al., 2021), MSCRED (Zhang et al., 2019), OmniAnomaly (Su et al., 2019), Time series foundation model Moment (Goswami et al., 2024), OneFitsAll (Zhou et al., 2023) and large language model GPT4 as our baselines. For more details, please refer to Appendix. A.1

**Datasets.** We use four real world datasets. Two of them are open source benchmarks: GECCO (Rehbach et al., 2018) and weather dataset (Institute, 2020). GECCO is a water quality monitoring dataset and weather is a weather monitoring dataset. Other two (Flight 1 and Flight 2) are aircraft operating system monitoring data of different aircrafts, collected from one of aircraft production company in World's top 300 enterprises. Among them, Flight 1 and Flight 2 have shown strong correlation lag across different dimensions. Part of weather dataset have shown weak correlation lag across different dimensions. GECCO does not show the correlation lag. Thus, we compare ADSec's performance on them in the following to verify the effectiveness of Neural Hawkes module. For more details of datasets, please refer to Appendix. A.2.

**Hyperparameter.** We use grid search to determine the optimal hyperparameter settings. We discuss the hyperparameter setting process, explore ranges, and optimal settings in Appendix. A.4.

**Evaluation Metrics.** We use three widely-used metrics: precision, recall and f1 score to measure the performance of ADSec and our baselines. For more details please refer to Appendix. A.3.

Table 1: The average performance of ADSec and baselines on four datasets. We use the first four characters to represent each method.

| | GECCO | | | Weather | | | Flight 1 | | | Flight 2 | | |
|---|---|---|---|---|---|---|---|---|---|---|---|---|
| | Prec | Rec | F1 | Prec | Rec | F1 | Prec | Rec | F1 | Prec | Rec | F1 |
| AutoF | 0.367 | 0.779 | 0.295 | 0.291 | 0.927 | 0.441 | 0.750 | 0.985 | 0.852 | 0.886 | 0.949 | 0.916 |
| Anoma | 0.438 | 0.872 | 0.553 | 0.849 | 0.815 | 0.821 | 0.129 | **1.000** | 0.229 | 0.127 | 0.939 | 0.223 |
| TranA | **0.995** | 0.892 | 0.930 | 0.820 | **0.988** | 0.894 | **1.000** | 0.425 | 0.597 | **1.000** | 0.424 | 0.596 |
| MSCRe | 0.874 | 0.888 | 0.874 | 0.825 | 0.920 | 0.869 | 0.885 | 0.746 | 0.810 | 0.952 | 0.798 | 0.868 |
| OmniA | 0.895 | 0.892 | 0.883 | 0.621 | 0.792 | 0.694 | 0.566 | 0.478 | 0.518 | 0.723 | 0.606 | 0.659 |
| Momen | 0.328 | **0.922** | 0.466 | 0.882 | 0.509 | 0.645 | 0.583 | 0.731 | 0.649 | 0.605 | 0.765 | 0.676 |
| OneFi | 0.621 | 0.792 | 0.637 | **0.988** | 0.190 | 0.319 | 0.857 | 0.358 | 0.505 | 0.310 | 0.133 | 0.186 |
| GPT4 | 0.131 | 0.477 | 0.192 | 0.290 | 0.915 | 0.441 | 0.113 | 0.896 | 0.200 | 0.142 | 0.606 | 0.229 |
| ADSec | 0.996 | 0.911 | **0.941** | 0.955 | 0.958 | **0.956** | 0.917 | 0.993 | **0.953** | 0.978 | **0.968** | **0.973** |

## 4.2 PREDICTION ACCURACY

We evaluate the average performance of ADSec and several baselines across four real-world datasets. The experimental results are summarized in Tab. 1, where the best scores are shown in bold and the second-best scores are underlined. For brevity, only the first five characters of each method's name are presented.

From Tab. 1, we observe that our method consistently achieves the highest F1 score across all four datasets, improving the F1 score by an average of 5.8% over the strongest baseline. Among the baselines, TranAD achieves good F1 scores on GECCO and Weather, whereas AutoFormer performs well on Flight 1 and Flight 2. We attribute this to the pronounced cross-dimension correlation lag in the Flight datasets: AutoFormer's auto-correlation mechanism explicitly models correlations across different lags, enabling it to better capture such dependencies. This also explains TranAD's performance drop on Flight 1 and Flight 2.

Moreover, ADSec's performance across the four datasets closely aligns with the strength of cross-dimension correlation lag observed in each dataset. Flight 1 and Flight 2 exhibit the strongest lag, and ADSec achieves higher F1 scores on these datasets than on the other two. In contrast, GECCO shows minimal cross-dimension lag, where ADSec's performance is comparatively lower. These findings validate the effectiveness of the Neural Hawkes process employed by ADSec.

### 4.3 MEMORY AND TIME OVERHEAD

We measure the memory and time overhead of ADSec and baselines on a Linux server with dual AMD EPYC 7T83 CPUs, eight NVIDIA RTX 4090 GPUs, and 503 GiB RAM. For performance tests, only a single RTX 4090 GPU was used. In Fig. 3(a), we present the training time, inference time, and memory overhead of the baselines and ADSec. For clarity, the memory overhead is divided by 10, and the time overhead of GPT-4 is also scaled down by a factor of 10 due to its large gap from other methods. As shown in the figure, compared with the most lightweight baseline, ADSec achieves higher accuracy while introducing only modest additional training and storage costs.

### 4.4 HYPERPARAMETER SENSITIVITY

We evaluated the F1 score of ADSec with respect to different values of $m$ and $\gamma$, where $m \in \{3, 5, 7, 9, 11\}$ and $\gamma \in \{0.1, 0.3, 0.5, 0.7, 0.9, 1.0\}$. Due to space limitations, we set $\gamma_1, \gamma_2, \gamma_3$ uniformly to $\gamma$. A more fine-grained parameter tuning is discussed in the Appendix A.4. The experimental results are shown in Fig. 3(b), from which we can observe that the effect of $m$ on ADSec is greater than that of $\gamma$. Specifically, the F1 score of ADSec first increases and then decreases as $m$ grows, reaching its optimum around $m = 5$. This observation aligns with the average cross-dimension correlation lag we measured in the Flight dataset (discussed in Appendix A.2). We attribute this fluctuation to the fact that when $m$ is too small, it fails to capture the lagged correlations across dimensions with large cross-dimension correlation lags. Conversely, when $m$ is too large, since most dimensions do not exhibit such long correlation lags, increasing $m$ substantially raises the number of trainable parameters, thereby complicating the training process and reducing the likelihood of converging to an optimal solution. When $m$ is fixed, the impact of $\gamma$ on ADSec is minimal: across the above range of $\gamma$, the variation in ADSec's F1 score does not exceed $3.5\%$. When $m$ and $\gamma$ vary jointly, the gap between the best and worst hyperparameter settings is within $8\%$ in terms of ADSec's F1 score. Therefore, the performance of ADSec is not sensitive to this set of hyperparameters and does not incur complicated deployment overhead.

In addition, we evaluated the variation of ADSec's F1 score when the learning rate and window length were set to $\{0.1, 0.01, 0.001, 0.0001, 5 \times 10^{-4}, 1 \times 10^{-4}\}$ and $\{3, 5, 7, 9, 11\}$, respectively, as shown in Fig. 3(c). When the learning rate is set to $0.1$, ADSec achieves its best performance; however, the model's performance becomes unstable across different window lengths due to the overly large step size. As the learning rate decreases to $0.01$ and below, the performance of the model becomes more stable across varying window lengths, with the best overall performance observed at a learning rate of $0.01$ (except $0.1$). Overall, ADSec's performance varies within $7\%$, when the learning rate and window length vary in the aforementioned ranges. Thus, ADSec is also insensitive to this group of hyperparameters.

### 4.5 EFFECTIVENESS OF EACH MODULE

To validate the effectiveness of the Semantical Topology, Neural Hawkes, and Topology Refinement modules, we first conducted ablation studies by removing each module in turn and comparing the resulting model performance with that of the complete ADSec. When ablating the Semantical Topology module, we replaced the computed word embedding correlation matrix with a random matrix. The corresponding results are reported in Tab. 2, where we use 'ST', 'NH', 'TR' to represent Semantical Topology, Neural Hawkes and Topology Refinement separately. From the table, we can observe that the complete ADSec consistently outperforms the variants with ablated modules. On the GECCO dataset, the performance gap between ADSec and the variant without the Neural Hawkes module is relatively small. This is mainly because, the phenomenon of cross-dimension correlation lag is not evident in GECCO (shown in Appendix. A.2).

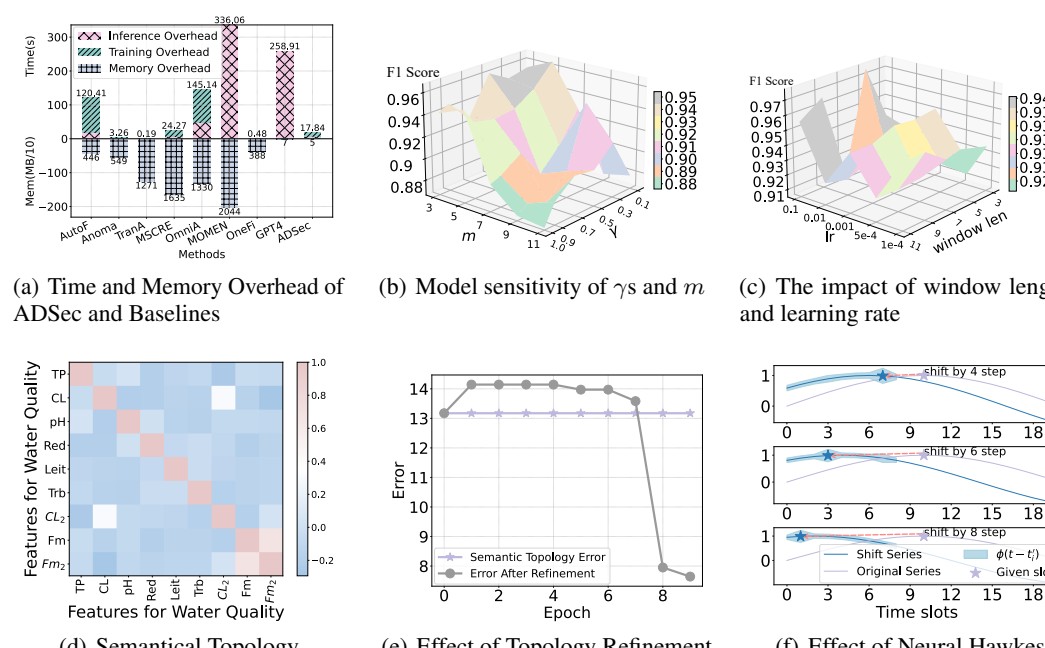

(a) Time and Memory Overhead of ADSec and Baselines

(b) Model sensitivity of $\gamma$s and $m$

(c) The impact of window length and learning rate

(d) Semantical Topology

(e) Effect of Topology Refinement

(f) Effect of Neural Hawkes

Figure 3: (a) The figure illustrates the memory overhead (divided by 10), as well as the inference and training time overhead of the baselines and ADSec. For brevity, we denote each method by its first five characters; (b)The figure shows the F1 scores of ADSec when $m$ and $\gamma_1$, $\gamma_2$, $\gamma_3$ change. For simplicity, we set $\gamma_1$ to $\gamma_3$ as a unified value $\gamma$. For more fine grained experiments, please refer to Appendix. A.4; (c) The figure shows the F1 scores of ADSec when window length and learning rate (lr) change; (d) The figure shows the correlation relationship between the dimensions of dataset GECCO. Positive value denotes positive correlation. The larger the value is, the stronger the correlation is and vice versa; (e) The figure shows the estimated error of semantical topology and the topology after refining along different training steps; (f) We shift one series to the left by 4, 6 and 8 steps and plot the $\phi(t - t'_i), t'_i \in \{t'_1, \ldots, t'_m\}$ of a given time slot $t$ in shadow. The wider the shadow is, the larger the value of $\phi(t - t'_i)$ is. The figure verifies that the point in the shifted series corresponding to the given time slot in original series has the largest $\phi(t - t'_i)$, which means the corresponding point can be attributed the highest weight in Neural Hawkes process.

Moreover, we exhibit the semantic topology in Fig. 3(d). The correlation results largely conform to water chemistry expectations, such as the negative association between temperature and redox, the inverse relation between $CL_2$ and pH, and the strong consistency between the two flow sensors. At the same time, certain deviations, like the unexpected negative correlation between $CL_2$ and Redox, indicate that semantic topology may be misleading. These findings highlight the necessity of data-driven topology refinement to uncover hidden or non-intuitive dependencies in multivariate time series.

We verify the effectiveness of the data-driven topology refinement module in Fig. 3(e), where we plot the estimated errors of the semantic topology and the refined topology. As shown, the estimated error of the refined topology first increases and then decreases as training proceeds. We attribute this behavior to the Adam optimizer: in the early stages, the moment estimates are not yet stable, leading to a temporary rise in error; once the estimates converge, the error begins to decrease.

To evaluate the effectiveness of the Neural Hawkes process, we construct synthetic datasets by shifting an original time series by 4, 6, and 8 steps to generate additional dimensions. As illustrated in Fig. 3(f), for a given time slot $t$, we highlight in the shifted series the corresponding values of $\phi(t - t'_i), i \in [1, m]$, using shaded regions whose widths increase with the magnitude of $\phi$. From the figure, it is evident that $\phi(t - t'_i)$ reaches its maximum at the point in the shifted series aligned with the given time slot. This demonstrates that the Neural Hawkes process is capable of capturing cross-dimension correlation lags.

Table 2: The average performance of ADSec and its variants.

| Modules | | | GECCO | | | Weather | | | Flight 1 | | | Flight 2 | | |
|---|---|---|---|---|---|---|---|---|---|---|---|---|---|---|
| ST | NH | TR | Prec | Rec | F1 | Prec | Rec | F1 | Prec | Rec | F1 | Prec | Rec | F1 |
| × | √ | √ | 0.873 | 0.860 | 0.863 | 0.882 | 0.959 | 0.918 | 0.709 | **1.000** | 0.830 | 0.857 | **0.968** | 0.909 |
| √ | × | √ | 0.962 | 0.911 | 0.924 | 0.879 | **0.985** | 0.928 | 0.796 | 0.993 | 0.884 | 0.657 | **0.968** | 0.783 |
| √ | √ | × | 0.892 | 0.839 | 0.853 | 0.857 | 0.955 | 0.898 | 0.624 | 0.694 | 0.657 | 0.891 | **0.968** | 0.928 |
| √ | √ | √ | **0.996** | 0.911 | **0.941** | **0.955** | 0.958 | **0.956** | **0.917** | 0.993 | **0.953** | **0.978** | 0.968 | **0.973** |

## 5 RELATED WORK

Graph neural networks can generally be divided into two categories: spectral methods and spatial methods. Spectral methods derive node representations based on graph spectral theory (Bruna et al., 2014; Defferrard et al., 2016; Kipf & Welling, 2017), whereas spatial methods perform graph convolutions directly in the spatial domain by aggregating information from neighboring nodes (Chen et al., 2018; Hamilton et al., 2017; Veličković et al., 2018). A limitation of these approaches is that they typically require a manually defined topology, which becomes cumbersome in high-dimensional settings. To address this issue, recent studies have proposed approaches that learn and refine the graph structure from training data (Jin et al., 2020; Wu et al., 2020). However, these methods usually assume that the graph structure remains static across time steps, and that a change in one dimension immediately leads to a change in another dimension, thereby employing a single graph throughout the entire time span. In reality, the relationships between different dimensions are often dynamic and exhibit cross-dimensional lags, as discussed in the Introduction. In other words, a change in one dimension at a given time may only induce a change in another dimension after several subsequent time steps. Consequently, no dependency exists at the time of the initial change, but a dependency emerges after a delay. Anomaly detection methods can broadly be categorized into classical approaches and neural network-based approaches. Classical methods, particularly statistical-based algorithms, model the distribution of time series data and identify anomalies according to their likelihood under the fitted distribution (Eskin, 2000; Wang et al., 2016). While effective for detecting point anomalies and distributional shifts, these methods typically rely on strong distributional assumptions that may not hold in dynamic environments (Ma et al., 2021). Moreover, they lack the capacity to incorporate expert knowledge and temporal dependencies, which limits their ability to detect contextual anomalies (Wang et al., 2024). Neural network-based methods, by contrast, can be further divided into prediction-based methods (Malhotra et al., 2015; Hundman et al., 2018; Zong et al., 2018; Chen et al., 2022a), reconstruction-based methods (Chen et al., 2022b; You et al., 2022; Jiang et al., 2022; Shen et al., 2021; Tian et al., 2019), and large language model (LLM)-based methods (Liu et al., 2024b; Russell-Gilbert et al., 2024; Liu et al., 2024a). However, prediction-based and reconstruction-based approaches often require manual effort to integrate expert knowledge and to construct appropriate graph structures. LLM-based methods, while more flexible, are generally insensitive to subtle value fluctuations and face challenges in capturing fine-grained normal patterns (Jin et al., 2024).

## 6 CONCLUSION

In this paper, we propose ADSec, a novel anomaly detection method that captures semantic topology and cross-dimensional correlation lag to better leverage expert knowledge and model dynamic inter-dimensional dependencies. ADSec has three key features: (1) it adopts a unified framework that integrates semantic topology extracted from expert documents with data-driven topology mined from monitoring data; (2) it employs a semantic topology extraction module to effectively utilize expert knowledge from diverse documents; and (3) it introduces a Neural Hawkes Process to capture cross-dimensional correlation lags and model dynamic topologies across different time intervals. Extensive experiments on four real-world datasets demonstrate that ADSec consistently improves F1 scores by an average of 5.8% over the strongest state-of-the-art anomaly detection methods, with each module contributing to this performance improvement.

## 7 REPRODUCIBILITY STATEMENT

We include the datasets and code used in our experiment in supplementary files. Besides, we discuss the hyperparameter tuning technique, LLM fine-tuning technique and optimal hyperparameter settings in Appendix. A.5 and Appendix. A.4.

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

## A   APPENDIX

### A.1   BASELINES

To verify the effectiveness of ADSec, we choose the SOTA anomaly detection methods: Anomaly Transformer, TranAD, OmniAnomaly, MSCRED, AutoFormer; SOTA time series foundation models: Moment, OneFitsAll and LLM-based method GPT4. Here is a brief introduction of each method.

- **Anomaly Transformer**: It introduces an association-based criterion for unsupervised time series anomaly detection. It employs a novel Anomaly-Attention mechanism to capture both local prior-associations and global series-associations, with a minimax strategy to enhance the distinction between normal and abnormal points. This design enables the model to achieve state-of-the-art performance across diverse anomaly detection benchmarks.

- **TranAD**: It is a transformer-based model for anomaly detection in multivariate time series. It uses a two-phase, self-conditioned reconstruction process with adversarial training to amplify subtle deviations and stabilize learning. Combined with meta-learning for data efficiency, TranAD achieves state-of-the-art detection accuracy while significantly reducing training and inference time.

- **OmniAnomaly**: It is a stochastic recurrent neural network designed for multivariate time series anomaly detection. It combines GRU with variational autoencoders and introduces techniques such as stochastic variable connection and planar normalizing flow to capture both temporal dependence and stochasticity of time series. By reconstructing inputs and evaluating reconstruction probabilities, it achieves robust anomaly detection and has shown superior performance across diverse real-world datasets.

- **MSCRED**: It is an unsupervised deep learning model for anomaly detection and diagnosis in multivariate time series. It constructs multi-scale signature matrices to represent system states, employs a convolutional encoder and attention-based ConvLSTM to capture inter-sensor correlations and temporal dependencies, and reconstructs these matrices with a convolutional decoder. Anomalies are identified through residuals, enabling both detection and root-cause diagnosis with improved robustness to noise.

- **AutoFormer**: AutoFormer is a Transformer-based model designed for long-term time series forecasting. It introduces a progressive decomposition architecture that separates trend and seasonal components during the forecasting process, thereby alleviating the challenges of modeling intricate temporal patterns. Moreover, AutoFormer replaces conventional self-attention with an Auto-Correlation mechanism, which leverages series periodicity to capture dependencies efficiently at the sub-series level, achieving both accuracy and scalability. In our experiments, we adopt AutoFormer within a prediction-based anomaly detection framework to perform anomaly detection.

- **Moment**: MOMENT is a family of open-source foundation models designed for general-purpose time series analysis. It is pre-trained on the large-scale and diverse Time Series Pile using masked time series modeling, enabling strong performance across forecasting, classification, anomaly detection, and imputation tasks. With minimal fine-tuning, MOMENT achieves competitive results in limited supervision settings, demonstrating its effectiveness as a versatile time series baseline.

- **OneFitsAll**: OneFitsAll leverages frozen pre-trained language or vision transformers for general time series analysis without modifying the core self-attention and feedforward layers. By fine-tuning only lightweight components, it provides a unified framework for diverse tasks including classification, forecasting, imputation, and anomaly detection. Experiments demonstrate that OneFitsAll achieves state-of-the-art or comparable performance across major time series benchmarks.

- **GPT4**: GPT-4 is a large-scale language model developed by OpenAI, known for its strong capabilities in natural language understanding and reasoning across diverse tasks. It has been widely applied in various domains due to its powerful generalization ability and adaptability. In our experiments, we employ GPT-4 with carefully designed prompt engineering to perform anomaly detection.

Table 3: Dataset characters

|  | GECCO | Weahter | Flight 1 | Flight 2 |
|---|---|---|---|---|
| Dimension | 9 | 19 | 179 | 179 |
| Anomaly Ratio | 5.40% | 26% | 11.90% | 10% |
| Stationary | Stable | Stable | Stable | Unstable |
| Periodicity | apeioridc | aperiodic | Weak periodicity | Weak periodicity |
| Dimension Correlation Lag | None | Little | Yes | Yes |
| Max Lag | – | 1 | 80 | 100 |
| Average Lag | – | 0.005 | 6.84 | 5.27 |

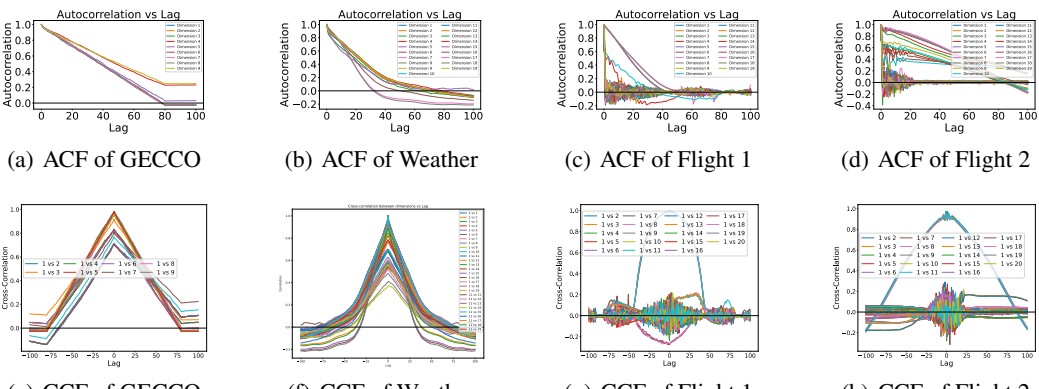

| (a) ACF of GECCO | (b) ACF of Weather | (c) ACF of Flight 1 | (d) ACF of Flight 2 |
|---|---|---|---|

| (e) CCF of GECCO | (f) CCF of Weather | (g) CCF of Flight 1 | (h) CCF of Flight 2 |
|---|---|---|---|

Figure 4: (a) The figure verifies that GECCO has no periodicity; (b) The figure verifies that Weather has no periodicity; (c) The figure verifies that Flight 1 has weak periodicity. Since the number of dimension in Flight 1 is too large, we only marked the first 10 dimension in legend; (d) The figure verifies that Flight 2 has weak periodicity; (e) The figure verifies that GECCO has no cross dimension correlation lag. Since the number of dimension in Flight 2 is too large, we only marked the first 10 dimension in legend; (f) The figure verifies that Weather has little cross dimension correlation lag; (g) The figure verifies that Flight 1 has lots of cross dimension correlation lag; (h) The figure verifies that Flight 2 has lots of cross dimension correlation lag.

## A.2 DATASETS

We use four real-world dataset: one water quality monitoring dataset, one weather dataset, two aircraft control system monitoring dataset. The GECCO dataset originates from the GECCO Industrial Challenge, focusing on online anomaly detection for drinking water quality. It contains real-world time series data collected from sensors monitoring water and environmental variables, provided by Thüringer Fernwasserversorgung, a major German water supplier. The dataset is designed to evaluate methods for detecting subtle but critical anomalies while maintaining low false alarm rates, making it a practical benchmark for robust anomaly detection research. The Weather dataset records 20 meteorological indicators every 10 minutes throughout 2020 at a Max Planck Institute station, covering parameters such as temperature, humidity, wind, radiation, and precipitation. With over 52,000 samples per variable, it provides high-resolution insights into atmospheric conditions and long-term weather patterns. For anomaly detection, we inserted synthetic anomalies into the original dataset to evaluate model performance under realistic perturbations. The Flight 1 and Flight 2 datasets are collected from the operation control systems of a top-300 global aircraft manufacturer. They include measurements from 179 different indicators. These datasets provide valuable benchmarks for evaluating anomaly detection methods in complex industrial systems.

Moreover, we analysis the stationary, periodicity, cross dimension correlation lag for each dataset, which is shown in Tab. 3. The Auto-Correlation Function (ACF) and Cross-Correlation Function (CCF) are shown in Fig. 4.

Table 4: Hyperparameter explore range

| Dataset | Hyperparameter | Ranges | Dataset | Hyperparameter | Ranges |
|---------|----------------|--------|---------|----------------|--------|
| GECCO | Learning rate | {0.1,0.01,**0.001**,0.005,**0.0001**} | Weather | Learning rate | {0.1,**0.01**,0.005,0.0001} |
| | Batch size | 100 | | Batch size | 100 |
| | m | {3,**5**,7,9,11} | | m | {3,**5**,7,9,11} |
| | $\gamma_1$ | {0.1,0.3,0.5,0.7,0.9,**1**} | | $\gamma_1$ | {0.1,0.3,0.5,0.7,0.9,**1**} |
| | $\gamma_2$ | {**0.1**,0.3,0.5,0.7,0.9,**1**} | | $\gamma_2$ | {**0.1**,0.3,0.5,0.7,0.9,**1**} |
| | $\gamma_3$ | {0.1,0.3,0.5,0.7,0.9,**1**} | | $\gamma_3$ | {0.1,0.3,0.5,0.7,0.9,**1**} |
| | Window length | {3,**5**,7,9,11} | | Window length | {3,**5**,7,9,11} |
| | Epochs | {**1**,**3**,5,**7**,**10**} | | Epochs | {1,**3**,5,**10**} |
| | Layer of attention | 10 | | Layer of attention | 10 |
| Flight1 | Learning rate | {0.1,0.01,0.005,**0.0001**} | Flight2 | Learning rate | {0.1,0.01,0.005,**0.0001**} |
| | Batch size | 100 | | Batch size | 100 |
| | m | {3,**5**,7,9,11} | | m | {3,**5**,7,9,11} |
| | $\gamma_1$ | {0.1,0.3,0.5,0.7,0.9,**1**} | | $\gamma_1$ | {0.1,0.3,0.5,0.7,0.9,**1**} |
| | $\gamma_2$ | {**0.1**,0.3,0.5,0.7,0.9,**1**} | | $\gamma_2$ | {**0.1**,0.3,0.5,0.7,0.9,**1**} |
| | $\gamma_3$ | {0.1,0.3,0.5,0.7,0.9,**1**} | | $\gamma_3$ | {0.1,0.3,0.5,0.7,0.9,**1**} |
| | Window length | {3,**5**,7,9,11} | | Window length | {3,**5**,7,9,11} |
| | Epochs | {**1**,3,5,10} | | Epochs | {**1**,3,5,10} |
| | Layer of attention | 10 | | Layer of attention | 10 |

## A.3 METRICS

In anomaly detection tasks, precision, recall, and F1 score are widely used to evaluate detection performance. Precision measures the proportion of correctly identified anomalies among all predicted anomalies, while recall reflects the proportion of true anomalies that are successfully detected. The F1 score is defined as the harmonic mean of $Precision$ and $Recall$. Formally, they are given by:

$$Precision = \frac{TP}{TP + FP} \tag{8}$$

$$Recall = \frac{TP}{TP + FN} \tag{9}$$

$$F1 = \frac{2 \times Precision \times Recall}{Precision + Recall} \tag{10}$$

where $TP$ denotes the number of correctly detected anomalies (true positives), $FP$ represents normal samples incorrectly predicted as anomalies (false positives), and $FN$ indicates anomalies that are missed by the detector (false negatives).

## A.4 HYPERPARAMETER SETTINGS

We use grid search to find the optimal hyperparameter settings for each dataset. We list the explore range and highlighted the optimal setting in Tab. 4. Moreover, we supplement more hyperparameter sensitivity experiments in Fig. 5, which verifies that ADSec is insensitive to different hyperparameter settings and requires reasonable deployment overhead.

## A.5 LLM TUNING TECHNIQUES

To enable efficient adaptation of Llama3-8B under limited hardware constraints, we employ parameter-efficient fine-tuning (PEFT) and model compression techniques.

4-bit quantization via the BitsAndBytes library reduces memory usage by approximately 75%, allowing the Llama3-8B model to be fine-tuned on GPUs with 10 GB of memory. We further apply Low-Rank Adaptation (LoRA), leveraging the low intrinsic rank of weight updates during adaptation. The rank of the low-rank matrices is set to 32, and the scaling factor lora_alpha is set to 64. LoRA is applied to the embedding layer to enhance word embeddings for downstream vector extraction tasks. The language modeling head (lm_head) is also fine-tuned to improve output accuracy. The resulting checkpoint size is typically only a few megabytes.

Optimization follows the causal language modeling (CLM) objective, enabling the model to capture syntactic and semantic patterns in meteorological text. We use the paged_adamw_8bit optimizer, which incorporates 8-bit quantization and paged memory to avoid out-of-memory errors during training.

A linear learning rate schedule decays the rate from 1e-5 to 4e-9 over 20 epochs, balancing rapid convergence with stable refinement. Training uses a per-device batch size of 2 with 2 gradient accumulation steps, yielding an effective batch size of 4. We use bfloat16 precision and gradient checkpointing, trading 20% additional computation for substantially reduced memory usage during backpropagation. Training proceeds for 20 epochs with metrics logged every 50 steps.

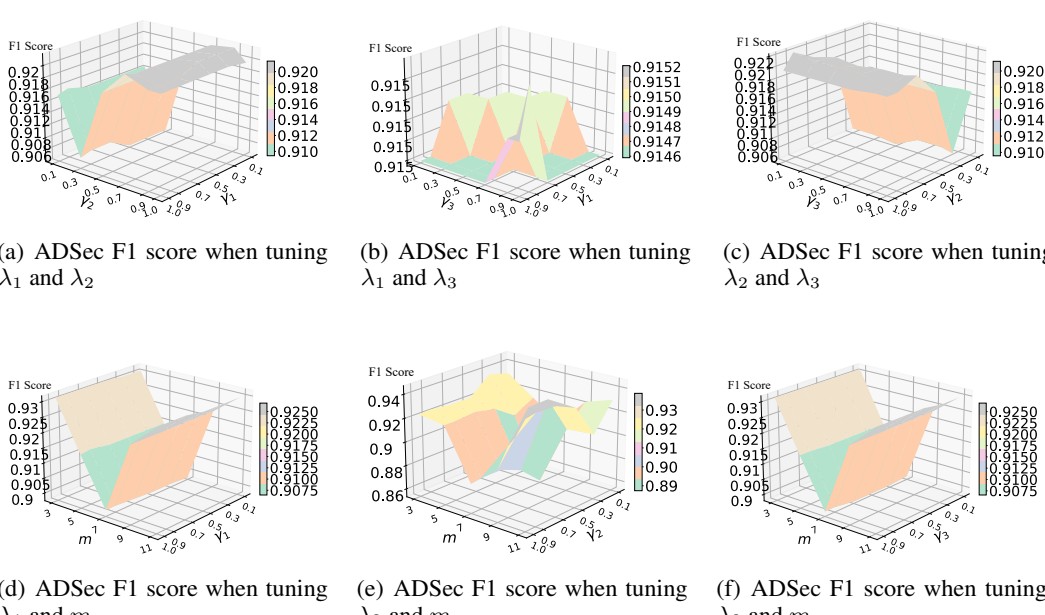

(a) ADSec F1 score when tuning $\lambda_1$ and $\lambda_2$

(b) ADSec F1 score when tuning $\lambda_1$ and $\lambda_3$

(c) ADSec F1 score when tuning $\lambda_2$ and $\lambda_3$

(d) ADSec F1 score when tuning $\lambda_1$ and $m$

(e) ADSec F1 score when tuning $\lambda_2$ and $m$

(f) ADSec F1 score when tuning $\lambda_3$ and $m$

Figure 5: (a-f) The figures show ADSec F1 score for different hyperparameter settings.

## A.6 USAGE OF LLM

In our paper writing process, we leverage large language models (LLMs) to assist with polishing and proofreading our English expressions. Moreover, during data visualization, we utilize LLMs to adjust figure legends, ticks, and other layout or formatting details.

