# OpenReview forum: "Semantic Embedding-Driven Topology Extraction with Correlation Lag Awareness"
_ICLR.cc/2026/Conference — ICLR 2026 Conference Withdrawn Submission_

### Official Review · Reviewer_xL2G · 2025-10-29

**Soundness:** 2
**Presentation:** 3
**Contribution:** 3
**Rating:** 4
**Confidence:** 4

**Summary:**

This paper introduces a novel anomaly detection method for multivariate time series with dynamic inter-dimensional dependencies. This method automatically extracts topology from expert documents using a fine-tuned LLM, and proposes a *Neural Hawkes Process*, which can capture cross-dimensional correlation lags, modeling time-shifted dependencies between different dimensions of the data. After Hawkes, they extracted the semantic topology, which was refined using monitoring data. The paper shows that ADSec improves performance in anomaly detection tasks by leveraging expert knowledge and dynamic dependencies, achieving a 5.8% improvement in F1 score compared to existing SOTA methods.

**Strengths:**

1. The integration of semantic topology from expert documents with data-driven refinement and Neural Hawkes Process to model dynamic time lags is a unique approach for anomaly detection in multivariate time series.
2. The paper evaluates ADSec on four real-world datasets, covering diverse scenarios from water quality monitoring to aircraft control systems, which demonstrates the generalizability of the method. It achieves superior performance on the evaluated datasets, outperforming SOTA methods. This is a key strength, especially in datasets with strong cross-dimensional correlation lags.

**Weaknesses:**

1. The paper does not provide error bars or conduct statistical significance tests like paired t-tests or confidence intervals. Without these statistical analyses, it is unclear whether the reported improvements are truly significant or merely due to variability in the experiments. Smaller performance improvements could be the result of experimental noise, rather than a real gain.
2. Could the authors provide concrete examples or case studies showing how semantic topology improves anomaly detection performance, particularly when compared to baseline methods that do not incorporate this expert knowledge?
3. The sensitivity analysis is conducted only on the Flight 1 dataset. I`m also curious about the overfitting of hyperparameters to that dataset, as hyperparameter choices might have different impacts on various datasets.
4. While this diversity of baselines provides a broad comparison and it has shown great behavior in capturing dependencies, it seems most baselines are not designed for capturing dynamic topologies. In fact, some SOTA unsupervised methods for anomaly detection in multivariate time series are designed for modeling dynamic graph relationships and get great performance, like MADGA[1] and MTGFlow[2]. So I recommend comparing ADSec with other graph-based and unsupervised methods, and it would be better if there are more case analyses between them.

[1]. Wang, Yuanyi, et al. "Interdependency matters: graph alignment for multivariate time series anomaly detection." 2024 IEEE International Conference on Data Mining (ICDM). IEEE, 2024.
[2]. Zhou, Qihang, et al. "Detecting multivariate time series anomalies with zero known label." Proceedings of the AAAI Conference on Artificial Intelligence. Vol. 37. No. 4. 2023.

**Questions:**

Please see the Weaknesses.

---

### Official Review · Reviewer_eMWX · 2025-10-31

**Soundness:** 3
**Presentation:** 3
**Contribution:** 3
**Rating:** 6
**Confidence:** 3

**Summary:**

Aiming at the core limitations of existing methods in multivariate time series anomaly detection (such as relying on manual topology construction and failing to model cross-dimensional correlation lags), this paper proposes ADSec (Anomaly Detection Method Capturing Semantic Topology and Correlation Lag). ADSec automatically extracts topology from expert documents and monitoring data, and models cross-dimensional correlation lags effectively: it first extracts semantic topology from expert documents and refines it with monitoring data, and then uses a novel Neural Hawkes process to model cross-dimensional correlation lags and adjust topology dynamically.

**Strengths:**

1. Multivariate time series anomaly detection has strong practical demands in fields such as industrial monitoring and environmental monitoring . However, high cost of manual topology construction and lack of dynamic lag modeling have long been persistent engineering and theoretical pain points in this field. ADSec proposes solutions to these two key pain points, and its problem positioning holds significant practical significance and academic value.
2. The "semantic topology extraction - dynamic lag modeling - topology refinement" framework proposed in the paper provides a new paradigm integrating expert knowledge and data-driven method for multivariate time series anomaly detection, which can inspire subsequent research.
3. ADSec demonstrates performance advantages on four real-world datasets, especially achieving an F1-score of 0.953–0.973 on aircraft datasets with strong lags. This proves that the method can be applied in high-complexity industrial scenarios. Meanwhile, the method’s low overhead and robustness provide feasibility for practical applications, and the results are of reference value to both industry and academia.

**Weaknesses:**

1. The limitations of the method have not been analyzed in the paper. For example, in ultra-large-scale scenarios where the number of dimensions exceeds 1000 (e.g., large-scale IoT systems), it is unclear whether the computational overhead of ADSec’s semantic topology extraction (LLM word embedding generation time) will increase significantly, and whether the alternating iteration of topology refinement will lead to excessively long training time.
2. Besides, the generalization of the method has not been verified enough. For instance, its performance on datasets from other domains  remains untested.

**Questions:**

1.	The semantic topology adjacency matrix is calculated using the cosine similarity of word embeddings . If two dimensions are semantically irrelevant but strongly correlated in data (such as "cabin temperature" and "fuel consumption" in aircraft data), will the semantic topology underestimate this correlation? How does the topology refinement module  balance semantic bias and data correlation? Can specific cases be provided ?
2.	The experiments only use four datasets, among which Flight 1 and Flight 2 are from the same aircraft manufacturer. How does ADSec perform on multivariate time series datasets in other fields (such as power load and traffic flow)?
3.	When the number of dimensions exceeds 1000 , what are the time/memory overheads of ADSec’s semantic topology extraction and topology refinement ? Are there targeted optimization strategies?

---

### Official Review · Reviewer_wpgx · 2025-10-31

**Soundness:** 2
**Presentation:** 2
**Contribution:** 2
**Rating:** 4
**Confidence:** 4

**Summary:**

This paper proposes an anomaly detection framework ADSec for multivariate time series that integrates semantic topology extraction from domain documents with data-driven temporal modeling. A fine-tuned LLM is utilized to derive semantic relations between variables. Moreover, the topology is refined using monitoring data, and the authors apply a Neural Hawkes process to capture dynamic cross-dimensional dependencies and correlation lags. Experiments on four real-world datasets show that ADSec improves F1 score by 5.8% over SOTA anomaly detection methods. Ablation studies confirm the contribution of each module.

**Strengths:**

S1. The paper tackles a practical issue that cross-dimensional correlation lag is underexplored in aircraft and IoT systems.

S2. The proposed architecture is clear, technically reasonable, and supported by ablation studies.

S3. Experiments on four real-world datasets show consistent improvement and robustness under hyperparameter variation.

**Weaknesses:**

W1. Lack of theoretical proof for the convergence and stability of the iterative optimization between the semantic and data-driven topology refinement.

W2. The proposed approach relies on the availability of high-quality expert documents and the feasibility of LLM fine-tuning, which is limited in environments lacking domain knowledge.

W3. The core modules in the architecture, such as semantic extraction and the Neural Hawkes process, are largely based on prior work, which has limited novelty.

W4. The paper lacks clear symbol definitions. Several sections contain symbols such as θ in section 3.4 and γ,γ_1,γ_2,γ_3 in sections 3.4 and 4.4 without explicit explanation, which makes the paper confusing to read.

W5. The reliability of the semantic topology is questionable. The authors mention that in Section 4.5, “the semantic topology may be misleading” (Fig.3(d)), and they attempt to address this issue through a data-driven topology refinement module. However, the reliability improvement is only empirically shown in Fig.3(e) without theoretical guarantees or robustness analysis.

**Questions:**

Please refer to the above weaknesses.

---

### Official Review · Reviewer_VzEJ · 2025-11-01

**Soundness:** 3
**Presentation:** 3
**Contribution:** 2
**Rating:** 4
**Confidence:** 3

**Summary:**

This paper introduces **ADSec** (Anomaly Detection method capturing Semantic Topology and Correlation lag), an approach for multivariate time series anomaly detection. The method addresses two key limitations in existing approaches: (1) the manual construction of topologies in graph-based methods, which is impractical for systems with thousands of indicators, and (2) the assumption of fixed, instantaneous dependencies that fails to capture cross-dimensional correlation lags.

**Strengths:**

- Combining semantic knowledge from documents with data-driven patterns is an elegant solution to the topology construction problem.
- The subword tokenization approach (Section 3.2) for dealing with different terms referring to the same indicator is clever and well-motivated with the "steering wheel control force" vs. "steering wheel command force" example.
- The Neural Hawkes process adaptation allowing negative impact functions and learning lag-specific impact functions is a genuine contribution to modeling cross-dimensional dependencies.
- Discussion of memory/time overhead (Figure 3a) and hyperparameter sensitivity.

**Weaknesses:**

- No analysis of convergence properties for the alternating optimization. And why cosine similarity of embeddings captures causal/functional relationships? No discussion of identifiability issues when learning both topology and impact functions simultaneously.
- Largest evaluation dataset (GECCO) has 179 dimensions, but method is motivated by **thousands of indicators**. Also, Weather dataset uses inserted synthetic anomalies, which may not reflect real anomaly patterns
- Missing comparisons with topology learning methods (DCRNN, GTA, MTGNN etc.)
- The smooth penalty term $\sum_{i,j} S_{i,j}|X[:,i] - X[:,j]|^2$ assumes smoother signals should be more connected, which may not hold for all systems
- Limited details on document preprocessing, quantity/quality requirements, and generalization to new domains
- No discussion of how anomaly thresholds are determined in practice

**Questions:**

1. How do you validate that the semantic topology $A$ actually captures meaningful relationships? Figure 3(d) shows some counter-intuitive correlations (CL2-Redox). What percentage of the semantic topology aligns with ground truth domain knowledge?
2. In Figure 3(e), why does the refined topology error initially increase? You attribute this to Adam optimizer momentum, but couldn't this indicate the semantic topology is being "unlearned" early in training? How do you ensure the refinement doesn't completely override valuable semantic information?
3. Table 2 shows removing Neural Hawkes (NH) has minimal impact on GECCO. Have you considered an adaptive approach that automatically determines when to use temporal lag modeling based on detected correlation lag characteristics?

---

### Note · Authors · 2025-11-26

**Comment:**

We sincerely thank the reviewers and the program committees for their time, constructive feedback, and thoughtful evaluation of our submission. After careful consideration, we believe that the current version of the paper still requires further adjustments and additional experiments to meet the standard we aim for. Therefore, we would like to withdraw the manuscript at this stage. We appreciate the reviewers’ efforts and look forward to resubmitting an improved version in the future.

**Withdrawal Confirmation:**

I have read and agree with the venue's withdrawal policy on behalf of myself and my co-authors.